# DISCO-BENCH: A CONTEXT-AWARE EVALUATION BENCHMARK FOR LANGUAGE MODELLING

## ABSTRACT

Modeling large contexts, especially linguistic phenomena that span beyond individual sentences, is a fundamental yet challenging aspect of natural language processing (NLP). However, existing evaluation benchmarks primarily focus on the evaluation of inter-sentence properties and overlook critical discourse phenomena that cross sentences. To bridge the gap, we propose Disco-Bench, a benchmark that can evaluate intra-sentence contextual properties across a diverse set of NLP tasks, covering understanding, translation, and generation. Disco-Bench consists of 9 document-level testsets in the literature domain, which contain rich discourse phenomena (e.g. cohesion and coherence) in Chinese and/or English. For linguistic analysis, we also design a diagnostic test suite to probe the extent to which the evaluated models have internalized contextual information. We totally evaluate 20 general-purpose and domain-specific models based on advanced pretraining architectures and large language models (LLMs). Our results show that (1) our evaluation benchmark is both challenging and necessary; (2) fine-grained pretraining with literary document-level training data consistently enhances the modeling of discourse information. We will release the datasets, pretrained models, and leaderboard, which we hope can significantly facilitate research in this field.

## 1 INTRODUCTION

To evaluate the general performance of models, previous work proposed a variety of benchmarks, covering different tasks and languages such as GLUE (Wang et al., 2018), CLUE (Xu et al., 2020) and XGLUE (Liang et al., 2020). However, existing benchmarks pay little attention to intra-sentence contextual properties such as discourse, which are fundamental and challenging problems in natural language processing (NLP) (Kevitt et al., 1992). A text generally consists of meaningful, unified, and purposive groups of sentences, which are organized as a whole (Cook, 1989). As shown in Figure 1, the

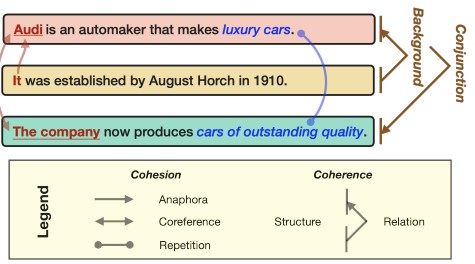

Figure 1: An example with intra-sentence contextual propertie.

discourse property manifests in two ways: (1) **cohesion**, where the dependency between words or phrases makes them logically and consistently connected; (2) **coherence**, where the structural relation between segments or sentences enables them semantically and meaningfully composed.

To bridge the gap, we introduce a novel benchmark for the target evaluation on the context-aware modeling. Our Disco-Bench comprises three datasets:

- **Disco-Bench Benchmark**: It consists of nine Chinese/English context-aware tasks covering a broad range of NLP tasks (understanding, translation, and generation), data quantities (from 26.4K to 2.4M), and difficulties. Besides, most task datasets are newly created in this work.
- **Disco-Bench Diagnostic Dataset**: To understand the discourse information learned by models, we propose a dataset of hand-crafted 1,294 examples for probing trained models. Each instance is a contrastive pair, where the correct candidate is the original instance in the benchmark and the incorrect one is a perturbation by modifying discourse devises in the correct candidates.

Table 1: An overview of our context-aware evaluation benchmark, covering language understanding, translation and generation. All datasets consist of document-level texts in the literature domain, which are rich in discourse phenomena. Eight of them are newly created by us and one is expanded based on existing corpus (i.e. MRC). It covers three languages: English (en), Modern Chinese (mzh/zh) and Classical Chinese (czh). We report commonly-used evaluation metrics. "#" means the number of instances (e.g. sentences, pairs or documents). "Test" represents both validation and testing sets.

| Task | Metric | Dataset | | | Language |
|------|--------|---------|---------|--------|----------|
| | | # Train | # Test | Domain | |
| *Understanding Task* | | | | | |
| SI | **F1, EM** | 48.0K | 17.5K | novel | zh |
| ZPR | **F1, P, R** | 2.2M | 8.1K | mixed | zh |
| MRC | **Acc.** | 26.4K | 6.5K | composition | mzh, czh |
| *Translation Task* | | | | | |
| NT | **d-BLEU,** | 1.9M | 1.3K | novel | zh→en |
| CCT | **BLEU, TER,** | 778.1K | 5.3K | dianji | czh→mzh |
| PT | **MET. COM.** | 47.1K | 2.7K | poetry | zh→en |
| *Generation Task* | | | | | |
| TE | **BLEU, PPL** | 2.4M | 10K | book | en |
| TI | **PPL, Dist,** | 233K | 10K | book | zh |
| TC | **BERTscore** | 233K | 10K | book | zh |

- **Disco-Bench Training Dataset**: We introduce a large-scale (400G), long-text data in Chinese and English, which is in the same literature domain with the benchmark. The training data enables fine-grained pretraining to better model context-aware information required by the benchmark.

To better understand challenges posed by Disco-Bench, we conduct experiments on a variety of state-of-the-art models, including standard Transformer, pretrained models as well as large language models (LLMs). We found that these tasks display different levels of difficulty, resulting in different behaviors and performances across models. Furthermore, the fine-grained pretraining based on the context-rich Disco-Bench training data improves performances particularly on cohesive translation and coherent generation. However, the best models still achieve a fairly low absolute score, highlighting the difficulty of modeling discourse. There are three **main contributions** in this work:

- **Challenging Tasks**: We propose a diverse set of context-aware tasks to evaluate monolingual and cross-lingual models' ability to understand, translate and generate texts.
- **Considerable Resources**: We build and release a variety of context-aware resources, including benchmarking datasets, diagnostic test suite and large-scale pretraining corpus.
- **Comprehensive Comparisons**: We systematically compare many advanced pretraining methods on the benchmark, and identify current challenges in context modelling for future exploration.

## 2 DISCO-BENCH BENCHMARK

To comprehensively evaluate the target models, Disco-Bench covers three types of NLP tasks, including language understanding, translation and generation. We design the benchmarking tasks using the following criteria: (1) our tasks should measure the ability of models to handle contextual information, thus we define related tasks at different levels of difficulty; (2) our datasets should contain rich discourse phenomena, thus we build document-level datasets with whole contexts extracted from literary texts. As shown in Table 1, we introduce 9 tasks containing corresponding datasets in Chinese and/or English: eight of which are newly created, and one is expanded based on existing data.

### 2.1 LANGUAGE UNDERSTANDING TASKS

Discourse is one of the fundamental problems for understanding models. It is difficult to determine the referents of pronouns and definite noun phrases, and understand elliptical sentence fragments, as well

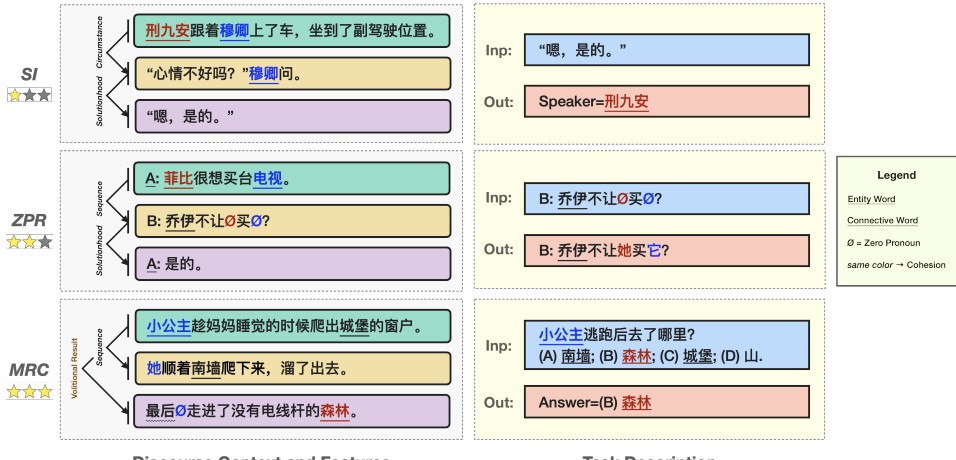

Figure 2: Illustration of the proposed **understating tasks** in terms discourse properties and task definition. As seen, SI needs to recognize named entity and resolve coreference. While ZPR demands the further ability to tackle zero anaphora and gender identification. MRC is the hardest because it should fully understand coherence (e.g. discourse structure based on temporal relation) apart from cohesion in previous tasks. English translations of example sentences are listed in Appendix §A.1.

as a host of other long-range language phenomena that have not even been adequately characterized much less conquered (Bates, 1995). As shown in Figure 2, we classify tasks into three difficulty levels according to the length of contexts and the amount of knowledge, required for discourse modeling.

**SI (Speaker Identification)**  Given a paragraph that may contain an utterance and the surrounding context, SI aims to identify the corresponding speaker(s) for the utterance or the content within quotation marks if no speaker exists. To archive this goal, models need to examine the existence of quotes, recognize named entities or phrases that can serve as speakers, and resolve coreference. We construct the dataset with 66K instances from eighteen Chinese novels. Unlike previous SI datasets like P&P (He et al., 2013) where all speakers are entities, speakers in our dataset can also be phrases, pronouns, or multi-entities. We employ macro-averaged F1 and exact match (EM) as the evaluation metrics, following standard extractive machine reading comprehension (Rajpurkar et al., 2016).

**ZPR (Zero Pronoun Recovery)**  ZPR aims to recover omitted pronouns in terms of position and form, according to its anaphora information in the given sentence (Yang & Xue, 2010; Zhang et al., 2019b; Song et al., 2020). Figure 2 shows an example, where the omitted pronoun "她 (She)" can be recovered according to its anaphora "菲比 (Phoebe)". The BaiduKnows is a widely-used Chinese ZPR corpus, which contains only 5K human-annotated sentences extracted from a Q&A forum (Zhang et al., 2019b). The insufficient data limits the investigation of model performance on ZPR. Inspired by Wang et al. (2016), we automatically built a large-scale training set from Chinese-English movie subtitles using word alignments. For testset, we hire experts to manually annotate 8K sentences covering five domains and the label set contains 30 Chinese pronouns. Different from previous benchmarks like CLUEWSC2020 which mainly focus on anaphora resolution (explicit pronouns) (Kong & Zhou, 2010; Mitkov, 2014), while ZPR considers implicit pronouns which are complementary to each other. We use micro F1, precision and recall as the evaluation metrics.

**MRC (Machine Reading Comprehension)**  The goal of MRC is to answer questions based on the understanding of its meaning given an unstructured text (Liu et al., 2019a; Zeng et al., 2020). We collected the Haihua2021 corpus, which contains 8K articles extracted from reading comprehension tests in primary/high school examinations.[1] Each article is followed by at least one question with 2∼5 choices and one correct answer. We manually create 2K articles as an additional supplement. Different from previous benchmarks based on Wikipedia texts (Cui et al., 2019) or Chinese idioms (Zheng et al., 2019), ours is in the literary domain (i.e. modern/ancient composition and poetry) that contains

---

[1] https://www.biendata.xyz/competition/haihua_2021.

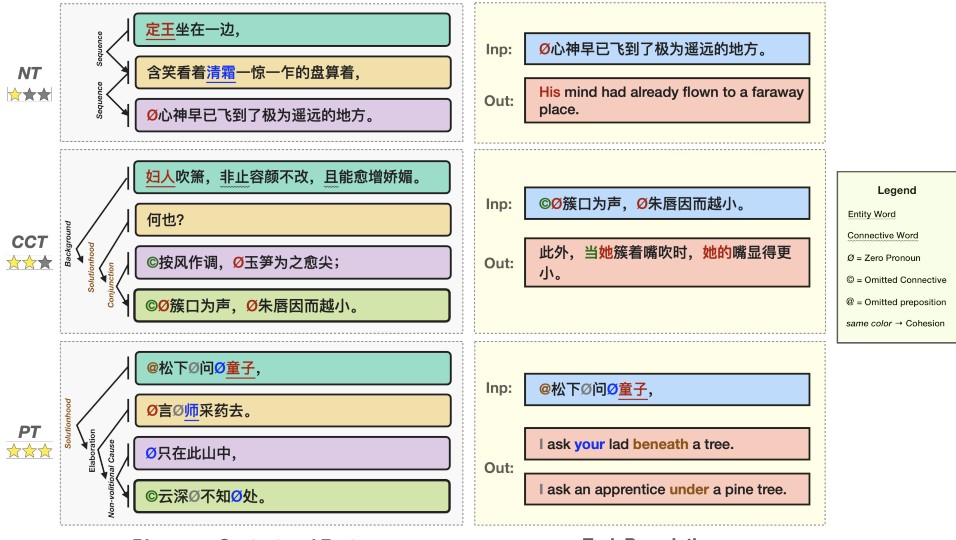

Figure 3: The illustration of the proposed **translation tasks** in terms of discourse properties and task definition. As seen, a variety of elements may be omitted in the Chinese input but should be recalled in English translation. NT mainly deals with zero pronouns while CCT needs to further tackle omitted connective words that are the marker of discourse structure. PT is the most difficult task because even prepositions could be further omitted. English translation is in Appendix §A.1.

rich discourse phenomena. Different from the $C^3$ benchmark (Sun et al., 2020) where problems are collected from Chinese-as-a-second-language examinations, this dataset is extracted from more challenging examinations designed for native speakers. Considering the average length of texts, our corpus is more challenging than $C^3$ (i.e. the length ratio is 753:117).

## 2.2 LANGUAGE TRANSLATION TASKS

Language translation is a sequence-to-sequence generation task to translate text from one language to another. Context information is important for document-level translation to produce cohesive and coherent translations (Wang et al., 2017; Bawden et al., 2018). As shown in Figure 3, we design three translation tasks of increasing hardness, which differ in the conciseness of source sentences in Chinese. The more concise the Chinese text, the more discourse information is needed for translation. We report BLEU, TER, METEOR and COMET for measuring models' translation quality.

**NT (Novel Translation)** The significant challenges for translating novels are entity consistency, anaphora resolution, and lexical choice (Matusov, 2019). We build a document-level Chinese-English corpus, which is extracted from web fictions. Specifically, we crawl 45,134 chapters in 152 books from web fiction websites, covering 14 genres such as fantasy science and romance. We manually align them at both document and sentence levels. Different from previous document-level MT datasets such as LDC[2] and OpenSubtitle[3] from the news and movie subtitle domains, ours is the first literature-domain MT corpus containing richer linguistic phenomena especially in discourse.

**CCT (Classical Chinese Translation)** Classical Chinese is a traditional style of written Chinese used in China until the early 20th century, making it different from any modern spoken form of Chinese. Compared with modern Chinese as in novel translation, classical Chinese texts are extremely concise and compact by often dropping subjects and objects when a reference to them is understood, which require discourse information for information recovery. We construct a document-level

---

[2]https://www.ldc.upenn.edu.
[3]https://opus.nlpl.eu/OpenSubtitles-v2018.php.

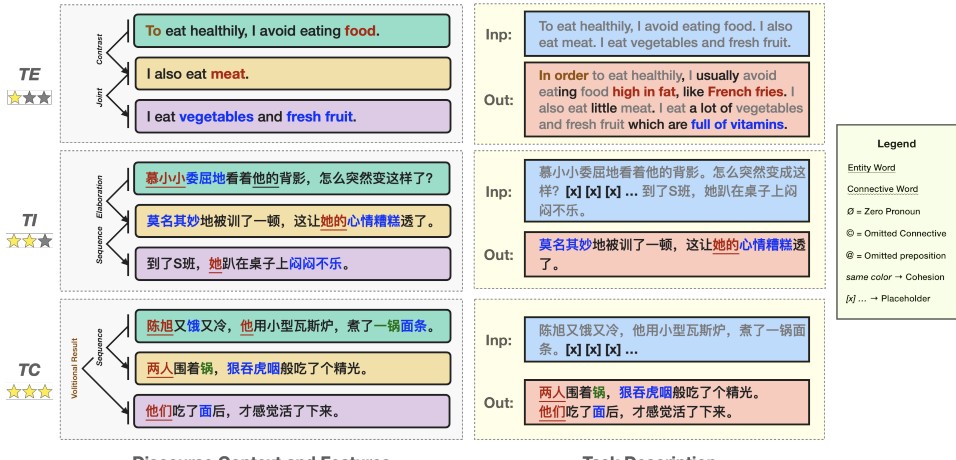

Figure 4: The illustration of the proposed **generation tasks** in terms of discourse properties and task definition. As seen, discourse structure and main contents have been specified in TE, thus the task needs to generate cohesive words. While TI should further consider cohesion relations when generating a whole sentence based on the previous and following ones. TC is the most difficult as it needs to generate more sentences with a unified structure. English translation is in Appendix §A.1.

Classical-Modern Chinese translation dataset, extracted from Chinese classics across history branch.[4] Different from the NiuTrans corpus[5] that has no context, ours maintain the original context.

**PT (Poetry Translation)**   Poetry translation is regarded as one of the hardest tasks in computational linguistics, or even artificial intelligence in general (Genzel et al., 2010; Ghazvininejad et al., 2018). Chinese poetry is even more concise than classic Chinese with implicit coherence, which is generally reflected through situational context and contextual context. For example, Chinese poetry does not use any cohesive means, but the semantic is still clear. We build a document-level Chinese Poetry to Modern English translation corpus, covering different types of Chinese poetry (e.g. Shi, Ci, Qu, and Fu) translated by famous translators.

## 2.3 LANGUAGE GENERATION TASKS

Language generation is a sequence generation task to produce text based on a given context (Reiter & Dale, 1997). Generating long and coherent text is an important but challenging task, particularly on lexical cohesion (Wanner, 1996; Guan et al., 2021). As shown in Figure 4, we design three representative generation tasks that differ in degrees of freedom. The more open-ended the generation task, the more difficult to generate accurate cohesive devices and discourse structure.

**TE (Text Expansion)**   We define a new task: given a predefined text, the goal of TE is to insert appropriate words, phrases, or clauses for adding more details and deepening the meaning, while retaining coherence and cohesiveness. We use a semi-automatic generation method to obtain large-scale training data. Specifically, we use the Stanford Parser[6] to produce the syntactic tree of a text, and then manually design some rules to delete the modifier words and phrases in the text. We use the remaining words as the input and predict the dropped modifier. Since some delete operations may produce ill-formed text, we filter out the training instances if the remaining text has a large perplexity measured by a language model. In order to retain the coherence and meaning of the source document, the expanded parts in the target text tends to be modifier phrases or clauses. We use BLEU and PPL metrics to measure the lexical and semantic similarities and fluency.

---

[4]https://en.wikipedia.org/wiki/Chinese_classics.
[5]https://github.com/NiuTrans/Classical-Modern.
[6]https://github.com/stanfordnlp/CoreNLP.

**TI (Text Infilling)**   It aims to predict a text snippet given its surrounding context (Zhu et al., 2019). To evaluate the discourse-level model capability, we focus on the sentence infilling task that predicts a missing bridge sentence $x_0$ given two preceding sentences ($x_{-2}$ and $x_{-1}$) and two subsequent sentences ($x_1$ and $x_2$) (Huang et al., 2020; Cai et al., 2020). We build a new TI dataset by extracting consecutive 5-sentence paragraphs from Chinese web fictions used in the NT task. To evaluate different models, we take the following automatic metrics: Perplexity (PPL), BLEU (Papineni et al., 2002), BERTscore (Zhang et al., 2019a) and diversity scores (Dist-2/4) (Li et al., 2016). We report degree of diversity by calculating the ratio of distinct 2-grams/4-grams in generated text.

**TC (Text Completion)**   The task is to predict a writing continuation given a preceding prompt. We focus on multi-sentence paragraph completion for a target evaluation of discourse modeling, which completes a multi-sentence paragraph $x_{s:e}$ given its leading sentence $x_s$. We use the same data collected for the TI task to construct the TC dataset. Specifically, given a sentence $x_{-2}$, we aim to predict the concatenation of $x_{-1}$, $x_0$, $x_1$, and $x_2$. We use the same metrics as TI task.

## 2.4   HUMAN EVALUATION ON BENCHMARK QUALITY

We assess the quality of our benchmark, as listed in Table 2. For the language understanding testsets that require human annotations, we follow Mitani et al. (2017) to calculate the inter-annotator agreement via Cohen's kappa (0∼1). The annotators reach high agreement on the testsets of understanding tasks, especially on the MRC testset, which annotates the correct answer from 2∼4 choices.

For translation and generation testsets, we randomly choose 100 instances for each task, and ask two human annotators to assess their quality in terms of fluency (1∼5) and adequacy/coherence (1∼5). We follow Kreutzer et al. (2018); Popovic (2021) to calculate inter-annotator agreement via Krippendorff's $\alpha$(0∼1) (Krippendorff, 2013). All outputs are fluent and highly correlated with the input sentences (i.e. > 4) with reasonable agreement, showing that our benchmark has high quality.

Table 2: Human evaluation on the benchmark quality. We also report the inter-annotator agreement (in bracket) for the translation and generation tasks.

| Task | Expert Evaluation | |
|---|---|---|
| | *Agreement* | |
| **SI** | 0.76 | |
| **ZPR** | 0.91 | |
| **MRC** | 0.97 | |
| | *Fluency* | *Adequacy* |
| **NT** | 4.9 (0.60) | 4.7 (0.78) |
| **CCT** | 4.9 (0.65) | 4.9 (0.55) |
| **PT** | 4.7 (0.63) | 4.4 (0.69) |
| | *Fluency* | *Adequacy* |
| **TE** | 4.0 (0.51) | 4.1 (0.51) |
| **TI** | 4.3 (0.63) | 4.4 (0.55) |
| **TC** | 4.3 (0.63) | 4.4 (0.55) |

## 3   DISCO-BENCH DIAGNOSTIC TEST SUITE

The general-purpose automatic metrics (e.g. BLEU and PPL) may be not sufficient to distinguish model performance in terms of discourse knowledge (Wong & Kit, 2012; Müller et al., 2018; Voita et al., 2018; 2019; Lin et al., 2011). To better measure the ability of models on discourse modeling, we handcraft a discourse-aware test suite that is complementary to general evaluation.

**Definition and Annotation**   We adapt the idea of contrastive testing in our approach (Bawden et al., 2018; Voita et al., 2019; Cai & Xiong, 2020; He et al., 2022). We craft a test suite that encompasses 6 cohesion properties (i.e. Repetition, Synonyms, Ellipsis, Substitution, Conjunction) for both English and Chinese languages. The detailed definition and examples are listed in Appendix §A.2.

**Contrastive Testing**   Table 3 provides examples of how we formulate contrastive pairs for different tasks. Each instance in our methodology comprises a contrastive pair, consisting of a correct and an incorrect input/hypothesis based on cohesion properties. The original content from the test set serves as the correct candidate, while we introduce variations by altering its discourse devices, creating the incorrect candidates. We select one representative task from each type of Disco-Bench Benchmark. Accordingly, we adopt diverse strategies which vary based on the location of modification:

- **MRC (Understanding)**: To generate an incorrect candidate, we introduce noise into the input, transforming it from $x$ to $x'$, while keeping the hypothesis $y$ constant. Thus, each instance contains

Table 3: The illustration of the proposed test suite. We design each contrastive instance with correct and incorrect discourse markers in terms of cohesion and coherence. Tested systems are asked to rank candidates according to their model scores.

| Type | Input | Hypothesis |
|------|-------|------------|
| | *Understanding Task: MRC (Machine Reading Comprehension)* | |
| Conj. | **Context**: 小公主 爬出 城堡。(*The little princess escaped from the castle.*) 
 **Correct**: 最后 她 躲进 了 森林。(*In the end she hid in the forest.*) 
 **Incorrect**: 然而 她 躲进 了 森林。(*However, she hid in the forest.*) | 小公主 逃 跑 后 去 了 哪 里 ？(*Where did the little princess go after she escaped?*) 
 (A) 南墙 (*Southern Wall*) 
 (B) 森林 (*Forest*) 
 (C) 城堡 (*Castle*) |
| | Ranking: Context + Correct/Incorrect → Hypothesis → Probability | |
| | *Translation Task: NT (Novel Translation)* | |
| Refe. | **Context**: 定王 含笑 看着 清霜。(*King Ding looked at Qingshuang with a smile.*) 
 **Current**: 他 觉得 清霜 很 滑稽。 | **Correct**: He thinks Qingshuang is funny. 
 **Incorrect**: She think the Qingshuang is funny. |
| | Ranking: Context + Current → Correct/Incorrect → Probability | |
| | *Generation Task: TC (Text Completion)* | |
| Repe. | **Context**: 叶远 的 右臂 融合了 洪荒龙骨。(*Ye Yuan's right arm fused with the primordial dragon bone.*) 
 **Correct**: 但 叶远 感觉 自己的 右臂 快要断了。(*But Ye Yuan felt as if his right arm was about to break.*) 
 **Incorrect**: 但 叶远 感觉 自己的 左手 快要断了。(*But Ye Yuan felt as if his left hand was about to break.*) | 这 一 拳 的 威力，实在 是 太 强了！(*The power of this punch is too strong!*) |
| | Ranking: Context + Correct/Incorrect + Hypothesis → Probability | |

a correct $(x, y)$ and an incorrect $(x', y)$ candidate. We then calculate the probability of the golden label by inputting these into the relevant models.

- **NT (Translation)**: We introduce noise into the target translation to generate an incorrect candidate, transitioning $y$ to $y'$, while the source input $x$ remains unaltered. Each instance hence contains a correct $(x, y)$ and an incorrect $(x, y')$ candidate. Given the input and hypothesis, we calculate the probability of the hypothesis sequence using a forced-decoding method.

- **TC (Generation)**: Similar to the MRC task, we introduce noise into the input while the hypothesis remains unchanged. By combining the input and hypothesis, we directly calculate the probability of the entire sequence.

In conclusion, we have annotated a total of 250 instances for the MRC task, 500 for the NT task, and 250 for the TC task, each marked with 6 different types of cohesion. Given each instance, we assess different models on their ability to rank the correct candidate higher than the incorrect one.

## 4 EXPERIMENTS

### 4.1 SETUP

**Plain Models** We use the Transformer (Vaswani et al., 2017) with *base* and *big* configurations as our plain models. We use the Adam optimizer with $\beta_1 = 0.9$ and $\beta_2 = 0.98$, and employed large batching Ott et al. (2018) for model training. We set the max learning rate to 0.0007 and warmup-steps to 16000. All the dropout probabilities are set to 0.3.

**Existing Pretrained Models** We systematically compare SOTA pretraining models on our constructed discourse-aware benchmark, including BERT (Devlin et al., 2019), RoBERTa (Cui et al., 2020), AnchiBERT (Tian et al., 2021), MengziBERT (Zhang et al., 2021), BART (Lewis et al., 2020; Shao et al., 2021), mBART (Liu et al., 2020), GPT2 (Radford et al., 2019; Zhao et al., 2019), T5 (Raffel et al., 2020; Zhao et al., 2019) and ChatGPT (Ouyang et al., 2022). We fine-tuned these public models on the corresponding datasets for downstream tasks. For translation tasks, we use BERT-based pretrained models (e.g. BERT, RoBERTa) to initialize the encoder of NMT models. We

Table 4: Performance of baseline models on Disco-Bench benchmark. A similar table is presented on the online platform. **Bold** denotes the best result in each column. SI and ZPR are measured by F1 while MRC by accuracy. We report BLEU for NT, CCT, PT and TE, and BERTscore for others.

| Model | Understanding | | | Translation | | | Generation | | |
|---|---|---|---|---|---|---|---|---|---|
| | SI$^\uparrow$ | ZPR$^\uparrow$ | MRC$^\uparrow$ | NT$^\uparrow$ | CCT$^\uparrow$ | PT$^\uparrow$ | TE$^\uparrow$ | TI$^\uparrow$ | TC$^\uparrow$ |
| *Plain Models* | | | | | | | | | |
| Transformer (base) | 9.1 | 10.8 | 38.2 | 22.1 | 32.5 | 4.3 | 24.9 | 58.1 | 58.2 |
| Transformer (big) | 4.4 | 11.1 | 38.7 | 22.5 | 33.5 | 4.3 | 29.6 | 58.5 | 59.9 |
| *Existing Pretrained Models* | | | | | | | | | |
| BERT (base) | 85.1 | 24.5 | 51.6 | 22.8 | 42.5 | 6.1 | - | - | - |
| AnchiBERT (base) | 81.3 | 23.2 | 46.3 | 22.1 | 42.6 | 6.1 | - | - | - |
| MengziBERT (base) | 86.9 | 31.5 | 51.0 | 21.2 | 42.3 | 5.5 | - | - | - |
| RoBERTa (base) | 86.3 | 28.5 | 51.0 | 21.9 | 42.3 | 5.8 | - | - | - |
| RoBERTa (large) | 88.7 | 33.0 | 55.9 | 20.8 | 44.2 | 5.7 | - | - | - |
| GPT-2 | - | - | - | - | - | - | 30.0 | 59.4 | 57.6 |
| BART (large) | 86.5 | 32.8 | 50.2 | 21.7 | 43.3 | 7.3 | 33.8 | 62.2 | 60.3 |
| mBART (CC25) | - | - | - | 24.0 | - | 12.6 | - | - | - |
| *Disco-Bench Pretrained Models* | | | | | | | | | |
| RoBERTa (base) | 87.7 | 31.2 | 50.0 | 22.8 | **46.6** | 6.6 | - | - | - |
| RoBERTa (large) | **89.6** | **34.3** | 56.7 | 21.6 | 44.0 | 7.2 | - | - | - |
| GPT-2 | - | - | - | - | - | - | 32.5 | 59.7 | 60.2 |
| BART (large) | 86.6 | 33.5 | 50.3 | 23.2 | 43.8 | 7.1 | **36.2** | **62.4** | **60.7** |
| mBART (CC25) | - | - | - | **24.3** | - | **13.9** | - | - | - |
| *Large Language Models* | | | | | | | | | |
| GPT-3.5 | 78.7 | 13.5 | 48.6 | 22.5 | 22.2 | 8.1 | 24.2 | 59.7 | 59.0 |
| GPT-4 | 84.9 | 9.7 | **63.2** | 24.0 | 27.6 | 9.1 | 27.1 | 60.4 | 59.6 |

choose the hyper-parameters based on the performance on the validation set for each model. We fine-tune each model twice and report the averaged test results. We use few-shot for tesing ChatGPT. The fine-tuning hyper-parameters and ChaGPT's instructions are detailed in Appendix §A.3.

**Disco-Bench Pretrained Models**   We present an extensive Disco-Bench training dataset (400GB), consisting of both Chinese and English texts, designed to align with the benchmark's literature domain. The frequencies and types of discourse phenomena vary in different domains (Yang et al., 2015), leading to differences in model behavior and quality across domains. However, most existing pretrained models are trained on non-literature data (e.g. Wikipedia). To fill the gap, we follow Wang et al. (2022) to train the existing pretraining models (*coarse-grained pretraining*) on our Disco-Bench training data (*fine-grained pretraining*) to enhance context modelling. Specifically, we use the existing pretrained models for weight initialization, and further train the models on the Disco-Bench training data with the same loss. More details on data and training settings are described in Appendix §A.4.

## 4.2   MAIN RESULTS

Table 4 lists the results on the proposed benchmarks, using main evaluation metrics (results on additional evaluation metrics are detailed in Appendix §A.5). Concerning the existing pretrained models, pretraining improves performance over plain models in all tasks, which is consistent with previous studies. These results validate that the proposed benchmarks are reasonable. We evaluated the encoder-only architecture on tasks involving comprehension and translation. We also assessed the decoder-only architecture on tasks requiring generation, and the encoder-decoder architecture on all tasks. The reason some architectures were not tested on certain tasks is due to our preliminary experiences showing subpar performance in those particular tasks.

Among the BERT variants with the base setting, AncientBERT trained on small-scale classical Chinese data outperforms other models on CCT and PT, demonstrating the necessity of bridging the domain gap. Enlarging the model capacity usually improves performance (e.g. RoBERTa from base to large setting). The GPT-2 model exhibits superior performance on TE and TI tasks compared

Table 5: Results of selected models on Disco-Bench cohesion test suit. We assess models on their ability to rank the correct candidate higher than the incorrect one according to model score. We report overall accuracy (%).

| Type | Models | Rep. | Syn. | Con. | Ref. | Sub. | Ell. |
|------|--------|------|------|------|------|------|------|
| *Understanding (MRC)* | RoBERTa (large) | 66.7 | 61.4 | **68.0** | **64.0** | 69.8 | 25.0 |
| | + Disco-Bench Pretrain | **68.8** | **66.3** | 63.4 | 58.3 | 59.5 | **62.5** |
| | GPT-3.5 | 27.1 | 38.6 | 33.5 | 25.8 | 49.2 | 12.5 |
| | GPT-4 | 31.3 | 24.1 | 21.0 | 21.6 | 39.7 | 25.0 |
| *Translation (NT)* | mBART (CC25) | 94.0 | 85.3 | 92.7 | 95.9 | 83.3 | **76.5** |
| | + Disco-Bench Pretrain | **96.0** | **88.2** | **95.0** | **96.7** | **86.7** | 76.5 |
| | GPT-3.5 | 32.0 | 59.4 | 24.4 | 26.0 | 44.8 | 37.3 |
| | GPT-4 | 62.0 | 85.3 | 45.1 | 71.6 | 58.6 | 41.2 |
| *Generation (TC)* | BART(large) | 89.5 | 60.0 | 91.4 | 81.9 | 50.0 | **61.9** |
| | + Disco-Bench Pretrain | **90.8** | **84.0** | **94.3** | **84.5** | **56.0** | 47.6 |
| | GPT-3.5 | 26.3 | 16.0 | 11.4 | 10.3 | 25.0 | 23.8 |
| | GPT-4 | 60.5 | 52.0 | 11.4 | 50.9 | 37.5 | 19.0 |

to the plain Transformer model, but its performance is inferior on the TC task. The BART model excels in all generation tasks, underscoring the efficacy of the encoder-decoder architecture in such tasks. Pre-training with multilingual data, such as in the mBART model, can yield a more substantial improvement in translation quality than BART, particularly evident in NT and PT tasks.

Clearly, fine-grained pretraining on Disco-Bench training data outperforms their coarse-grained counterparts, demonstrating the effectiveness and necessity of modeling discourse information. The RoBERTa models work better on language understanding tasks, and the BART variants produce superior performances on the language translation and generation tasks. Although ChatGPT has shown substantial proficiency in long-text NLP tasks, it does not quite measure up to the performance of Disco-Bench's pretrained models across the majority of Disco-Bench tasks. These results underline the challenge and the necessity of our proposed benchmark.

### 4.3 Results on Diagnostic Test Suite

We evaluate three existing pretraining models on the diagnostic dataset: RoBERTa (large), BART (large), and mBART (CC25), each of which has exhibited superior performance on their respective representative tasks. "+ Disco-Bench Pretrain" donates fine-grained pretraining on Disco-Bench data specific to each model. Subsequently, every model is fine-tuned using the training data derived from the corresponding downstream task.

Table 5 records the model's ability to rank a correct candidate higher than an incorrect one, revealing an overall accuracy percentage. Disco-Bench pretrained models generally improve the cohesion accuracies over their coarse-grained counterparts, which reconfirms our claim that fine-grained pretraining on Disco-Bench data helps model discourse information. Although the numbers are not comparable across tasks, we find that pretraining models on the understanding tasks generally perform worse on discourse modeling. One possible reason is that the understanding tasks are mostly classification tasks, whose signals may not be sufficient to guide models to learn discourse information. The results on GPT-3.5 and GPT-4 reveal a significant performance gap between LLMs and those pretrained with Disco-Bench data, emphasizing the challenge of capturing discourse information.

## 5 Conclusion

This paper introduces a benchmark for Chinese and/or English that can evaluate intra-sentence properties across various NLP tasks, covering understanding, translation, and generation. We also propose a diagnostic test suite that can examine whether the target models learn discourse knowledge for in-depth linguistic analysis. Extensive experiments demonstrate that fine-grained pretraining based on document-level training data consistently improves the modeling of discourse information. We offer the datasets, pretrained models, and leaderboards to facilitate research in this field.

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

## A APPENDIX

### A.1 ENGLISH TRANSLATION OF FIGURE 4-6

Table 6 presents English translations of the examples from Figures 2, 3, and 4. Each row details the discourse context and the task description for a specific task. By mapping these discourse phenomena into English, we can better understand the tasks and their associated challenges when developing and evaluating models.

Table 6: English translations of examples in Figure 2, 3 and 4. Some are literal translations in order to map discourse phenomena into the English language.

| Task | Discourse Context | Task Description |
|---|---|---|
| | *Figure 2* | |
| SI | Xing Jiu'an followed Mu Qing into the car and sat in the co-pilot position. 
 "Are you in a bad mood?" Mu Qing asked. 
 "Um, yes." | Inp: "Um, yes." 
 Out: Speaker=Xing Jiu'an |
| ZPR | A: Phoebe would love to buy a TV. 
 B: Joey won't let ∅ buy ∅? 
 A: Yes. | Inp: B: Joey won't let ∅ buy ∅? 
 Out: B: Joey won't let her buy it? |
| MRC | The little princess climbed out of the castle window while her mother was sleeping. 
 She climbed down the south wall and slipped out. 
 Finally ∅ walked into the forest without telegraph poles. | Inp: Where did the little princess go after she escaped? 
 (A) South Wall; (B) Forest; (C) Castle; (D) Mountain. 
 Out: Answer=(B) Forest |
| | *Figure 3* | |
| NT | King Ding sat on the side, 
 smiling as he looked at Qing Shuang's astounded thoughts. 
 ∅ mind had already flown to a faraway place. | Inp: ∅ mind had already flown to a faraway place. 
 Out: – |
| CCT | ©, when she is playing Xiao, not only can her beautiful face remain as usual, but also her charm increases. Why? 
 © ∅ is playing, ∅ fingers press the holes on the flute, and in this way, ∅ tender and slim fingers will seem to be slimmer and fairer. 
 ©, when shrinking ∅ month to blow, ∅ mouth appears to be smaller. | Inp: ©, when shrinking ∅ month to blow, ∅ mouth appears to be smaller. 
 Out: Besides, when shrinking her month to blow, her mouth appears to be smaller. |
| PT | I ask your lad beneath a tree. 
 "My master's gone for herbs, " says he, 
 "Amid the hills I know not where, 
 For clouds have veiled them here and there. " | Inp: I ask your lad beneath a tree. 
 Out: – |
| | *Figure 4* | |
| TE | – | – |
| TI | Mu Xiaoxiao looked at his back aggrieved, why did it suddenly change like this? 
 She was inexplicably trained for a while, which made her feel bad. 
 When she got to class S, she was lying on the table and was sullen. | Inp: Mu Xiaoxiao looked at his back aggrieved, why did it suddenly change like this? [x] [x] [x] ... When she got to class S, she was lying on the table and was sullen. 
 Out: She was inexplicably trained for a while, which made her feel bad. |
| TC | Chen Xu was hungry and cold. He used a small gas stove to cook a pot of noodles. 
 The two gathered around the pot and devoured everything. 
 After they ate the noodles, they felt alive. | Inp: Chen Xu was hungry and cold. [x] [x] [x] ... 
 Out: The two gathered around the pot and devoured everything. After they ate the noodles, they felt alive. |

### A.2 DETAILS OF DIAGNOSTIC TEST SUITE

**Definition and Annotation** As shown in Table 7, we define 6 properties in our test suite:

- **Repetition** means the repeating of certain words or phrases. We mainly annotate nouns repetition in 4~5 neighbouring sentences.

- **Synonyms** means related words that having the same connotations, implications, or reference in two sentences. In our test suite, this phenomenon include nouns and adjectives synonyms in 4∼5 neighbouring sentences.
- **Ellipsis** means the omission of one or more words that are obviously understood but that must be supplied to make a construction grammatically complete. This omission often happens after wh-words in English and in subject elements in Chinese.
- **Substitution** occurs when one item within a text or discourse is replaced by another. In English, such nouns are often replaced by "one" or "some", and verbs are replaced by "do" or "did". In Chinese, this often happens around quantifier or temporal adverbial.
- **Reference** is a relationship between objects in which one object designates, or acts as a means by which to connect to or link to, another object.
- **Conjunction** expresses a logical semantic relationship between two sentences rather than between words or structures. We mainly annotate additive, adversative, causal, and temporal.

Table 7: Chinese Examples of cohesion phenomena in our test suite.

| Type | Example | Contrastive Instance |
|------|---------|---------------------|
| **Repetition** | 佑哥 独自 返身 又来到 拍卖行 ... 离开 拍卖行 后 佑哥 联系了 韩家公子 等人 ...
(*Youge went alone and returned to the auction house ... After leaving the auction house, Youge contacted the son of the Han family and others ...*) | 拍 卖 行 → [公 司\|家 里\|...]
(*auction house → [company\|home\|...]*) |
| **Synonyms** | 高手 不一定 要 很 英俊 ... 你 不要 看到 一个 长得 帅 的 法师 就说 是 ...
(*A master does not necessarily have to be very handsome ... Don't say a good-looking wizard is... when you see one.*) | 帅 → [丑陋\|怪异\| ...]
(*good-looking → [ugly\|weird\|...]*) |
| **Ellipsis** | 刘甲 问道: "不知 你们 要买 多少亩 水田 呢? " ... 连芳洲 就 笑笑，说道: "大概 两千来亩 ∅ 吧！" ...
(*Liu Jia asked, "I don't know how many acres of paddy fields you want to buy?" ... Lian Fangzhou just smiled and said, "About two thousand acres of ∅!"*) | ∅ → 水田
(*∅ → paddy fields*) |
| **Substitution** | 周二晚九时 我们 见到了 迈克。当时 我们 邀请 他出席 那个 晚会。
(*We met Mike at nine o'clock on Tuesday evening. At that time, we invited him to the party.*) | 当时 → 周二晚九时
(*At that time → nine o'clock on Tuesday evening*) |
| **Reference** | 不过 大卫 却是 毛骨悚然 ... 他 立即 停止 了 想 说出 更 多 ...
(*However, David was horrified ... He immediately stopped wanting to say more ...*) | 他 → [她\|它\|...]
(*He → [She\|It\|...]*) |
| **Conjunction** | 陈旭 心里 有些 疑, ... 不过，这时候 出言 顶撞，显然 是 不明智的。
(*Chen Xu was somewhat doubtful, ... however, it was obviously unwise to contradict at this time.*) | 不过 → [除非\|所以\|...]
(*However → [Unless\|Therefore\|...]*) |

**ChatGPT's Prompts for Diagnostic Testing**  Table 9 showcases the prompts used in the LLMs probing for the Disco-Bench Test Suit. Each row describes a specific task, such as Speaker Identification (SI), Zero Pronoun Recovery (ZPR), and Multiple-choice Reading Comprehension (MRC), along with their corresponding prompts. The prompts were designed to assess various aspects of language understanding, including context interpretation, anaphora resolution, translation, and text completion. For translation and text evaluation tasks, the LLMs are required to choose from multiple candidates, making these tasks challenging and comprehensive. The diagnostic prompts aid in benchmarking the performance of LLMs in various discourse-level tasks, and they serve as a resource to assess the coherence and cohesion understanding of the models.

## A.3 DETAILS OF EXISTING PRETRAINED MODELS

We evaluate the following public pretrained models on Disco-Bench Benchmark and Test Suite:

- **BERT (base)**: we use the base model (12 layer encoder, hidden size 768, vocabulary size 21128) published by Devlin et al. (2019), which was pretrained on Chinese Wikipedia dump of about 0.4 billion tokens using the losses of mask language model (MLM) and next sentence prediction.[7]
- **RoBERTa (base)**: Cui et al. (2020) a model with the same architecture of BERT (base) except it uses whole word masking and is trained on additional 5 billion tokens with only MLM pretrained task. This model uses BERT (base) as the initial weight.[8]
- **RoBERTa (large)**: Cui et al. (2020) the large model size of RoBERTa model (24 layer encoder, hidden size 1024, vocabulary size 21128) This model has the same training procedure of RoBERTa-wwm-ext (base). This model is trained from scratch.[9]
- **AnchiBERT**: Tian et al. (2021) a model continues pretraining based on the BERT (base) model with the 39.5M ancient Chinese tokens. It uses the same tokenizer and other techniques as BERT-base.[10]
- **MengziBERT**: Zhang et al. (2021) a model initial on the RoBERTa (base) (Liu et al., 2019b) with special-designed objectives.[11]
- **BART (large)**: Shao et al. (2021) train a large model (12 layer encoder and 12 layer decoder, hidden size 1024, vocabulary size 21128) with denoising auto-encoding (DAE) objective. This model is trained on the open source large-scale raw text, Chinese Wikipedia, and a part of WuDaoCorpus. The training data contains 200GB cleaned text ranging from different domains.[12]
- **mBART (CC25)**: Pires et al. (2019) use a large model (12 layer encoder and 12 layer decoder, hidden size 1024, vocabulary size 250,000), trained with 25 language web corpus. This model is trained from scratch.[13]
- **GPT2**: Zhao et al. (2019) train a 12-layer decoder-only Transformers and its vocabulary is size 21,128. This model is trained with the CLUECorpusSmall corpus.[14]
- **GPT-3.5 & GPT-4**: ChatGPTis an intelligent chatting machine developed by OpenAI upon the InstructGPT (Ouyang et al., 2022), which is trained to follow an instruction in a prompt and provide a detailed response. All corresponding results were obtained from ChatGPT API in June 2023.[15]

The fine-tuning hyper-parameters are detailed in Table 8. Table 9 showcases the ChatGPT's prompts used for the Disco-Bench Benchmark tasks.

Table 8: A summary of hyper-parameter for fine-tuning downstream tasks.

| Task | Batch Size | Max Length | Epoch | Learning Rate |
|------|------------|------------|-------|---------------|
| SI   | 64         | 512        | 5     | 3e-5          |
| ZPR  | 5          | 512        | 40    | 5e-6          |
| MRC  | 6          | 512        | 10    | 2e-5          |
| NT   | 3K token   | 1024       | 30K step | 1e-4       |
| ACT  | 3K token   | 1024       | 30K step | 1e-4       |
| PT   | 3K token   | 1024       | 30K step | 1e-5       |
| TE   | 32         | 512        | 3     | 2e-4          |
| TI   | 24         | 64         | 3     | 2e-5          |
| TC   | 24         | 512        | 8     | 2e-5          |

---

[7] https://huggingface.co/bert-base-chinese.

[8] https://huggingface.co/hfl/chinese-roberta-wwm-ext/tree/main.

[9] https://huggingface.co/hfl/chinese-roberta-wwm-ext.

[10] https://github.com/ttzHome/AnchiBERT.

[11] https://huggingface.co/Langboat/mengzi-bert-base.

[12] https://huggingface.co/fnlp/bart-base-chinese.

[13] https://dl.fbaipublicfiles.com/fairseq/models/mbart/mbart.cc25.v2.tar.gz

[14] https://github.com/CLUEbenchmark/CLUECorpus2020.

[15] https://platform.openai.com.

Table 9: The prompt for evaluating ChatGPT. $C$ represents the context for machine reading, $SRC$ and $TGT$ denote source and target languages, respectively. $D$ represents a document contains several sentences. $T_1 \ldots T_m$ refer to the translation candidates, where only one of them is a positive translation and the others are negative due to the modification of discourse-specific words.

| Task | Prompt |
|------|--------|
| | *Disco-Bench Benchmark* |
| SI | In this cloze reading comprehension task, I will input a passage of text and a sentence, and you will need to find relevant information from the text and determine the speaker of the sentence. Passage: $P$, Question: $Q$, Speaker: |
| ZPR | The zero-anaphora recovery task is to restore the expression of omitted pronouns in terms of position and form based on the anaphoric information in the sentence. Please restore the original sentence with <> as the marker. If there is no zero-anaphora phenomenon, output "none." |
| MRC | Answer the following multiple-choice questions. Choose $A, B, C, or D$ as the final answer. "Content": $C$, "Question": $Q$, "Choices": $[C_1 C_2 C_3 C_4]$, "Answer": |
| NT | Translate the given Chinese into English. $D$ |
| CCT | Translate this ancient text into modern Chinese. $D$ |
| PT | Translate the given Chinese into English. $D$ |
| TE | given a predefined text, the goal of TE is to insert appropriate words, phrases, or clauses for adding more details and deepening the meaning, while retaining coherence and cohesiveness." $D$ |
| TI | The purpose of the text filling task is to predict text fragments based on context. The input includes the two sentences before and after the target sentence. Please output the target sentence. $S_{-2}, S_{-1}, S_1, S_2$ |
| TC | Based on the given context, the text completion task requires outputting the next four sentences. $S_{-2}$ |
| | *Disco-Bench Cohesion Test Suit* |
| MRC | Output the model's confidence for the answer based on the content and corresponding answer of the following multiple-choice reading comprehension. Answer the confidence for the following multiple-choice questions. Choose A, B, C, or D as the final answer. "Content": C, "Question": Q,"Choices": $[C_1 C_2 C_3 C_4]$,"Answer": "$C_x$", "Confidence": |
| NT | According to the Chinese text, which of the following is the correct English translation? Please output the correct translation's corresponding number. Chinese: D English:$[T_1, T_2, ..., T_m]$. Correct translation number: |
| TC | Given the Chinese text, please evaluate the following sentences based on cohesion and fluency, and output the corresponding number of the optimal sentences: $[S_1, S_2, ..., S_m]$. |

## A.4 DETAILS OF DISCO-BENCH PRETRAINED MODELS

**Disco-Bench Training Data** As shown in Table 10, this corpus includes numerous categories, such as Electronic, Modernist, Ancient, and Others, each further divided into specific genres. For the Chinese language, we offer millions of documents ranging from web fiction to ancient texts. For the English language, the dataset includes a similarly wide range, from web fiction to classical masterpieces and beyond. Overall, this rich dataset provides a thorough foundation for training sophisticated language models, emphasizing the fine-grained understanding of discourse information.

Comparing our corpus to other commonly used datasets for pretraining models, Disco-Bench's dataset exhibits distinct attributes and advantages (as shown in Table 11). Most of the currently available corpora, such as the Wikipedia used for Chinese BERT (base), have limited data size, approximately 1.5GB. The multilingual datasets, such as those for BART (large) and mBART (CC25), incorporate Chinese, English, and more languages. However, even though they present a larger size (200GB and 1.4TB respectively), their sources are often confined to Wikipedia, WuDao Corpus, or Common Crawl. In summary, the Disco-Bench dataset excels in terms of language diversity, corpus size, and the uniqueness of data sources, marking it as a valuable resource for diverse and comprehensive language model pretraining.

Table 10: Statistics of data for Disco-Bench pretraining. All data are extracted from literature texts with discourse context. We count number of characters in Chinese and number of words in English.

| Category | Genre | Size | | | Description |
|---|---|---|---|---|---|
| | | # Document | # Sentence | # Chara./Word | |
| *Chinese Language* | | | | | |
| Electronic | Novel | 91,620,211 | 1,169,127,191 | 58,639,454,317 | Web Fiction |
| Modernist | Classical | 38,495,887 | 490,733,235 | 24,613,514,541 | Masterpiece |
| | Book | 324,912 | 4,141,874 | 155,189,807 | Publication |
| Ancient | Poetry | 378,323 | 1,495,466 | 31,746,541 | Shi, Ci, Qu, Fu |
| | Couplet | 8,979,186 | 8,979,186 | 192,214,600 | Antithetical Couplet |
| | Classical | 1,011 | 1,947,136 | 53,721,504 | Ancient Text |
| Others | Lyrics | 452,715 | 4,952,039 | 165,338,679 | World's Songs |
| | Screenplay | 5,213 | 10,426,213 | 156,390,000 | Movie Script |
| | Movie | 66,050 | 24,108,241 | 642,392,397 | Movie Subtitle |
| | Dialogue | 3,642 | 1,653,469 | 49,406,618 | Talk, Message |
| Total | | 140,327,150 | 1,717,564,050 | 84,699,369,004 | |
| *English Language* | | | | | |
| Electronic | Novel | 33,156,134 | 422,757,234 | 26,777,401,794 | Web Fiction |
| Modernist | Classical | 3,104,507 | 39,593,119 | 2,507,247,359 | Masterpiece |
| | Book | 324,912 | 4,162,821 | 78,695,499 | Publication |
| Ancient | Poetry | 2,269 | 21,456 | 148,222 | World's Poetry |
| Others | Lyrics | 3,088,688 | 110,268,328 | 632,820,393 | World's Songs |
| | Movie Script | 2,826 | 12,534,815 | 67,433,609 | Movie Script |
| | Movie | 155,670 | 56,819,567 | 315,189,001 | Movie Subtitle |
| | Dialogue | 9,191 | 4,172,736 | 27,208,957 | Talk, Message |
| Total | | 39,844,197 | 650,330,076 | 30,406,144,834 | |

Table 11: Summary of pretrained models varying in model architecture, parameter scale, training data, and targeted task (i.e. understanding, translation, and generation). #1∼11 are publicly available. #12 denote a series of pretrained models that are continuously trained on our literature-domain data initialized by corresponding parameters in #1∼11.

| # | Model | Language | Size | Task | Corpus | |
|---|---|---|---|---|---|---|
| | | | | | Size | Sources |
| 1 | BERT (base) | zh | 110M | U, T | 1.5GB | Wiki |
| 2 | RoBERTa (base) | zh | 110M | U, T | 15GB | Wiki, EXT Corpus |
| 3 | RoBERTa (large) | zh | 340M | U, T | 15GB | Wiki, EXT Corpus |
| 4 | AnchiBERT (base) | zh | 102M | U, T | 1.5GB | Classical Chinese |
| 5 | MengziBERT (base) | zh | 103M | U, T | 300GB | Wiki, Common Crawl |
| 6 | BART (large) | zh, en | 406M | U, T, G | 200GB | Wiki, WuDao Corpus |
| 7 | mBART (CC25) | zh, en, etc. | 610M | T | 1.4TB | Common Crawl |
| 8 | GPT2 (base) | zh | 102M | G | 14GB | CLEU Corpus |
| 9 | GPT2 (large) | en | 762M | G | 40GB | Web Text |
| 10 | T5 (base) | zh | 231M | G | 14GB | CLEU Corpus |
| 11 | T5 (large) | en | 770M | G | 745GB | C4 |
| 12 | Disco-Bench (family) | zh, en | – | U, T, G | 400GB | Literature |

**Fine-grained Pretraining with Disco-Bench Training Data**  The pretraining hyper-parameters details of the Disco-Bench models can be found in Table 12.

Table 12: The summary of hyper-parameters used for Disco-Bench pretrained models.

| Model | RoBERTa | GPT2 | BART | mBART |
|---|---|---|---|---|
| Tokenization | BERTtok. | BERTtok. | BERTtok. | SentPiece |
| Optimizer | Adam | Adam | Adam | Adam |
| Masking | word | - | word | word |
| Vocabulary Size | 21128 | 21131 | 21128 | 250000 |
| Learning Rate | 3e-4 | 3e-4 | 3e-4 | 3e-4 |
| Batch Size | 4K | 4K | 4K | 4K |
| Training Step | 1M | 1M | 1M | 1M |
| Max Length | 512 | 1024 | 512 | 1024 |
| Layer | 12/24 | 20 | 24 | 12/24 |
| Head | 12/16 | 36 | 16 | 12/16 |
| Total Param. | 110m/340m | 737M | 406M | 669M |

## A.5 RESULTS ON ADDITIONAL EVALUATION METRICS

A single automatic evaluation metric might not provide a comprehensive depiction of a model's performance. We report the results on several additional evaluation metrics.

**Understanding Tasks** Table 13 presents additional evaluation metrics for understanding tasks, including Exact Match (whether the system's response exactly matches the correct answer) for SI and both Precision (how many of the predicted positive responses were actually positive) and Recall (how many of the actual positive responses were correctly identified by the system) for ZPR. The performance of the Disco-Bench pretrained RoBERTa (large) model according to additional metrics is consistently superior and comparable to the other models. This corroborates our conclusions drawn from the main evaluation metrics. Notably, the existing pretrained RoBERTa (large) model shows the highest Precision at 39.3 on the ZPR task.

Table 13: More results on **understanding tasks** using additional evaluation metrics, including Exact Match, Precision, and Recall. This is complementary to Table 4.

| Model | SI | ZPR | |
|---|---|---|---|
| | Exact Match$^\uparrow$ | Precision$^\uparrow$ | Recall$^\uparrow$ |
| *Plain Models* | | | |
| Transformer (base) | 0.3 | 10.2 | 11.5 |
| Transformer (big) | 0.1 | 10.5 | 11.9 |
| *Existing Pretrained Models* | | | |
| BERT (base) | 81.9 | 26.1 | 31.0 |
| AnchiBERT | 76.9 | 22.1 | 24.6 |
| MengziBERT | 84.0 | 36.6 | 29.6 |
| RoBERTa (base) | 83.4 | 29.0 | 29.9 |
| RoBERTa (large) | 85.9 | **39.3** | 28.7 |
| BART (large) | 83.7 | 38.3 | 30.2 |
| *Disco-Bench Pretrained Models* | | | |
| RoBERTa (base) | 85.2 | 32.0 | 30.6 |
| RoBERTa (large) | **87.2** | 38.7 | **30.8** |
| BART (large) | 84.6 | 39.0 | 30.5 |

**Translation Tasks** Table 14 provides supplementary evaluation metrics for translation tasks, comprising TER (measuring the number of edits required to change a system's output into one of the references), METEOR (considering precision and recall, synonymy, stemming, and phrase-level matches to create an F-score-like composite of these factors), and COMET (learned metric trained on human translation ranking data, which captures more nuanced, semantic comparisons and is less reliant on surface-level text matches). Notably, there are no resources available for Classical Chinese in the METEOR evaluation. When observing the performance across NT and PT tasks, the

Disco-Bench pretrained mBART model outshines all others across all three metrics, reinforcing its top-ranking performance as indicated by the BLEU scores. However, the metrics TER and COMET display inconsistent performances when applied to the CCT task, thereby illustrating the inherent challenges in evaluating such tasks.

Table 14: More results on **translation tasks** using additional evaluation metrics, including TER, METEOR and COMET. This is complementary to Table 4.

| Model | NT | | | CCT | | PT | | |
|---|---|---|---|---|---|---|---|---|
| | TER$\downarrow$ | MET.$\uparrow$ | COM.$\uparrow$ | TER$\downarrow$ | COM.$\uparrow$ | TER$\downarrow$ | MET.$\uparrow$ | COM.$\uparrow$ |
| *Plain Models* | | | | | | | | |
| Transformer (base) | 74.3 | 20.6 | 0.74 | 98.5 | 0.65 | 114.1 | 7.4 | 0.48 |
| Transformer (big) | 73.3 | 20.9 | 0.75 | 98.4 | 0.65 | 112.9 | 7.9 | 0.49 |
| *Existing Pretrained Models* | | | | | | | | |
| BERT (base) | 73.7 | 21.1 | 0.74 | 95.8 | 0.65 | 105.9 | 10.4 | 0.52 |
| AnchiBERT | 74.1 | 20.7 | 0.74 | 95.9 | 0.67 | 100.1 | 10.4 | 0.53 |
| MengziBERT | 76.5 | 20.5 | 0.74 | 96.0 | 0.67 | 105.5 | 8.9 | 0.51 |
| RoBERTa (base) | 74.1 | 20.5 | 0.75 | 96.2 | 0.65 | 104.7 | 9.1 | 0.51 |
| RoBERTa (large) | 75.1 | 19.6 | 0.72 | 94.8 | 0.68 | 99.6 | 9.4 | 0.50 |
| BART (large) | 75.6 | 21.1 | 0.74 | 96.5 | 0.65 | 100.8 | 11.1 | 0.54 |
| mBART(CC25) | 71.9 | 22.2 | 0.77 | - | - | 88.2 | 14.7 | **0.64** |
| *Disco-Bench Pretrained Models* | | | | | | | | |
| RoBERTa (base) | 73.6 | 21.0 | 0.75 | **91.5** | 0.67 | 104.1 | 9.3 | 0.51 |
| RoBERTa (large) | 74.6 | 20.5 | 0.75 | 95.5 | 0.67 | 102.0 | 9.6 | 0.51 |
| BART (large) | 72.0 | 21.2 | 0.76 | 96.7 | **0.70** | 100.0 | 12.0 | 0.57 |
| mBART (large) | **70.8** | **22.8** | **0.78** | - | - | **84.6** | **14.9** | **0.64** |

**Generation Tasks**   Table 15 introduces additional evaluation metrics for generation tasks, comprising PPL [16] (perplexity is a measurement of how well a probability distribution or probability model predicts a sample), BLEU (evaluating the quality of text which has been machine-generated based on reference), and Dist-$n$ (calculating the number of unique n-grams divided by the total number of n-grams in the generated text). As seen these metrics exhibit varying performances, highlighting the complexities and challenges associated with the automatic evaluation of generation tasks. Dist-2 and Dist-4 exhibit consistent performance in line with the primary metric, BERTscore. Conversely, the performances of PPL and BLEU metrics are notably unstable.

Table 15: More results on **generation tasks** using additional evaluation metrics, including BLEU, PPL, Dist-2 and Dist-4. This is complementary to Table 4.

| Model | TE | TI | | | | TC | | | |
|---|---|---|---|---|---|---|---|---|---|
| | PPL$\downarrow$ | BLEU$\uparrow$ | PPL$\downarrow$ | Dist-2$\uparrow$ | Dist-4$\uparrow$ | BLEU$\uparrow$ | PPL$\downarrow$ | Dist-2$\uparrow$ | Dist-4$\uparrow$ |
| *Existing Pretrained Models* | | | | | | | | | |
| BART (large) | 63.1 | **3.7** | **8.4** | 0.20 | 0.63 | 2.7 | 3.8 | 0.07 | 0.42 |
| GPT-2 | 70.1 | 1.6 | 11.2 | 0.18 | 0.54 | 2.1 | **2.7** | 0.03 | 0.17 |
| *Disco-Bench Pretrained Models* | | | | | | | | | |
| BART (large) | **49.2** | **3.7** | 8.8 | 0.19 | 0.65 | 2.9 | 3.3 | 0.05 | 0.29 |
| GPT-2 | 67.5 | 2.2 | 11.5 | **0.27** | **0.84** | **4.7** | 3.9 | **0.08** | **0.51** |

## A.6   RELATED WORK

Evaluation benchmarks are important for developing deep learning models, which enable comparison between different models and probe models for understanding of specific linguistic phenomena. Conneau & Kiela (2018) collected SentEval containing several sentence-level classification tasks

---

[16]We use GPT2 language model to compute PPL. For TI and TE tasks, we use 'IDEA-CCNL/Wenzhong-GPT2-110M'.

to test the representational power of models. Closely related to this work, DiscoEval (Chen et al., 2019) extended these tasks to evaluate discourse-related knowledge in pretrained models. DiscoEval only evaluates sentence encoder with language understanding tasks in English. In contrast, we extend the tasks to a boarder range of NLP tasks, which can evaluate different types of models (e.g. encoder-based BERT, decoder-based GPT, and encoder-decoder based mBART). In addition, our benchmarks cover both Chinese and English.

GLUE (Wang et al., 2018) and SuperGLUE (Wang et al., 2019) included a wider variety of natural language understanding tasks, further examining the capabilities of the models and making the results comparable for multi-task learning. Followed researchers extend the benchmarks to other languages, such as CLUE (Xu et al., 2020) and LOT (Guan et al., 2022) in Chinese, and XGLUE (Liang et al., 2020) in multiple languages. While these works focus on evaluating inter-sentence information,[17] our benchmark evaluates intra-sentence discourse phenomena that cross sentences.

---

[17]LOT (Guan et al., 2022) evaluates models' abilities to model long text but ignores discourse information.

