# OpenReview forum: "Disco-Bench: A Context-Aware Evaluation Benchmark for Language Modelling"
_ICLR.cc/2024/Conference — Submitted to ICLR 2024_

### Official Review · Reviewer_8L44 · 2023-10-30

**Soundness:** 2 fair
**Presentation:** 2 fair
**Contribution:** 1 poor
**Rating:** 3
**Confidence:** 4

**Summary:**

NLP test suites like Glue or Super Glue have been used for evaluating LM capabilities. This paper presents a new benchmark containing various NLP tasks, as well as a set of diagnostic probes. The two main claims to novelty by the authors are that : 1) This dataset contains both english and chinese sentences so it generalizes beyond just one language. 2) This diagnostic dataset portion of their benchmark requires models to understand discourse phenomena. The discourse understanding is quantified by measuring models along metrics such as Repetition, identifying Ellipsis, identifying substitutions.

**Strengths:**

This paper presents a new benchmark dataset and presents experiments using various LLMs on this dataset.

**Weaknesses:**

The benchmark tasks do not seem to be very novel and different from benchmarks such as Super Glue or LOT. The performance of existing pretrained Roberta (large) on the undertanding tasks is (88.7, 33.0, 55.9) which does not seem be very far from the performance of Disco-Bench pretrained models.

**Questions:**

The majority of the "Disco-Bench benchmark" contains tasks such as translation, reading comprehension, question answering, and text completion, text infilling which are already covered by various other NLP benchmarks/datasets. The novelty in the paper seems to be the hand-crafted probe sentences in the diagnostic test suite that encompasses 6 cohesion properties .

Why are these cohesion properties chosen ? Why are they important ?

What portion of the performance difference in Table 5 b/w vanilla models, and disco-bench pretrained models comes simply from domain/language mismatch ?

---

> ### Author Response · Authors · 2023-11-22
> **Response to Reviewer 8L44**
>
> **Q1. About two viewpoints in Weaknesses.**
>
> It appears that the motivation and primary contributions of our paper may not been fully grasped:
> 1. Table 2 in the Super GLUE benchmark explicitly demonstrates that the majority of tasks are classification-based comprehension tasks [1]. In contrast, our work additionally encompasses context-aware translation and generation tasks, offering a broader scope.
> 2. The objective of this paper is not to introduce a superior model; instead, the Disco-Bench pretrained models are designed as robust baseline models for conducting diversity analysis. Therefore, it is inappropriate for the reviewer to list these viewpoints as weaknesses.
>
> [1] SuperGLUE: A Stickier Benchmark for General-Purpose Language Understanding Systems.
>
> **Q2. Why are these cohesion properties chosen? Why are they important?**
>
> The related contents are introduced on discourse property and Disco-Bench Diagnostic Dataset in Introduction Section, first two paragraphs in Section 3 and detailed references.
>
> A text generally consists of meaningful, unified, and purposive groups of sentences, which are organized as a whole. As shown in Figure 1, the discourse property manifests in two ways: (1) cohesion, where the dependency between words or phrases makes them logically and consistently connected; (2) coherence, where the structural relation between segments or sentences enables them semantically and meaningfully composed. To understand the discourse information learned by models, we propose a dataset of hand-crafted contrastive pair for probing trained models.
>
> **Q3. What portion of the performance difference in Table 5 comes simply from domain/language mismatch?**
> 1. The difference in model training lies solely in the incorporation of disco-bench pretraining data during the pretraining phase, while a same downstream task data is used during the finetuning stage.
> 2. Discourse phenomena are present in texts across all domains, but literary texts contain richer discourse elements. Our aim in the pretraining phase is to imbibe more discourse knowledge.
> 3. Discourse knowledge often exhibits similarities across various languages. This suggests that simply increasing the data volume for a single language may not necessarily boost the model's capability in processing discourse for that specific language. Rather, the comprehension and management of discourse are predominantly cross-linguistic in nature.
>
> Hence, our dataset is crafted to evaluate the model's overarching ability in handling discourse-level information, rather than being confined to a specific domain/language.

---

### Official Review · Reviewer_maWf · 2023-11-01

**Soundness:** 3 good
**Presentation:** 1 poor
**Contribution:** 3 good
**Rating:** 5
**Confidence:** 2

**Summary:**

This presents 9 large discourse datasets with a focus on Chinese data (modern or classical), Chinese-English paired data, and one English task, alongside a diagnostics suite.

It is very ambitious in not simply bundling 9 existing datasets but actually providing 9 new datasets, as well as a large unannotated data resource (2+ billion sentences). They train models on the raw data and their 9 tasks and provide evaluations using a range of these models, and they provide essentially a tenth task/dataset for diagnostic tests using perturbations of discourse markers.

**Strengths:**

- It is very ambitious in not simply bundling 9 existing datasets but actually providing 9 new datasets.
- They seem to provide new coverage of the literary domain across a range of tasks.
- They provide newly collected document-level MT datasets which, if of sufficient quality, seem like they would contribute great value.
- They provide essentially a tenth task/dataset for diagnostic tests

**Weaknesses:**

- I would have appreciated far more detail regarding how these datasets are annotated, and the paper lacks analysis to measure their quality.
- For almost every dataset, there is an existing dataset on that task and this paper proposes a new version of the task, without providing the advantages of their new task or enough details to compare them. That issue is particularly fraught for ZPR, where the benchmark proposes to replace a manually annotated dataset with an automatically generated one. The same issue arises with a lack of reasons provided for using this dataset in comparison to existing long-form Chinese benchmark (Guan et al. 2022)
- A number of these datasets dismiss relevant smaller high-quality datasets in exchange for automatically generated tasks, particularly in the context of the ZPR data. But there's simply no real experimental analysis that actually establishes the validity of making those judgement calls, or the quality of their automatically bootstrapped datasets

**Questions:**

I appreciate that the IAA scores are provided for ZP recovery, but I'm confused about the numbers: the ZPR has kappa in the 0.9+ (implying a very clear-cut task), but the best model performance of 34.3 in this data -- compared to 46.81 F1 model performance for other ZP resolution models from years ago , which are presumably the same task (Song et al. (2020) -- https://aclanthology.org/2020.acl-main.482.pdf). Do the authors have a suggestion for the reason for the dramatic difference in behavior across different Chinese ZP recovery tasks? Is the domain chosen particularly hard?
What are the stars to the left of each figure?
Table 1 caption says that"" means the number of instances (e.g. sentences, pairs or documents)." -- what are the units for SI, ZPR and MRC? Quotations, implied pronouns, and questions, I assume?
I was confused at the actual evaluation of the diagnostic task: are models simply evaluated regarding whether they assign higher probability to the correct discourse marker?  Why do the authors not use this task to evaluate any models?
The paper says their new MRC dataset "contains rich discourse phenomena", and is from "more challenging" domain, but doesn't provide any detail beyond document length. Could they provide any examples or analysis?
Similarly, they claim the NT task contains "richer linguistic phenomena, especially in discourse.", but again provide no detail: could they explain what they mean?
The authors claim that their "Novel Translation" dataset took 5,134 chapters in 152 books (Table 3 suggests it's 1.4 million units -- I assume sentences?) and they manually aligned it. Does that mean that the authors did millions of alignment annotations? If not, could they clarify their methodology?

---

> ### Author Response · Authors · 2023-11-22
> **Response to Reviewer maWf**
>
> **Q1. All details about ZPR.**
>
> I apologize for any confusion caused by the lack of information. In the revised version, within the Appendix, we will provide comprehensive background knowledge and references for each task.
> 1. **About replacing a manually annotated dataset with an automatically generated one.** First, the newly introduced 8.1K testing set remains manually annotated and spans five distinct text domains  (but Zhang et al., 2019b only contains 1K single domain testing set). Second, we introduced an alternative, automatically generated training set, which significantly addresses the issue of long-standing data scarcity (but Zhang et al., 2019b only contains 5K human-annotated training set). Such kind of auto-annotated ZP training datasets have been thoroughly validated across various tasks, including ZPR, MT, question answering, and summarization tasks [1-3]. For future work, there is the flexibility to opt for manually annotated, automatically annotated, or a hybrid dataset for fine-tuning. Our approach simply offers more options in this regard.
> 2. **High IAA scores yet low model performance & different scores in Song et al. (2020).** Professionally trained annotators (we employed university professors specializing in linguistics and translation studies), having received thorough annotation training, are indeed capable of achieving a high level of consistency in their annotations. While such omissions do not pose a problem for humans, as they can easily recall missing pronouns from the context, the task remains a significant challenge for models. This discrepancy is precisely why we focus on this task in the benchmark. Song et al. (2020) only use BaiduZhidao QA forum training and testing set (Zhang et al., 2019b), however, we report socres averaged on large five domains (movie subtitle, QA forum, government news, web fiction, and personal profile).
> 3. **Difficulty stars in Figure 1~3.** As stated in figure captions and first graph of each subsetion in Section 2, the stars denote difficulty levels of different tasks. These levels are determined based on key factors: 1) the length of contexts required to find cues, and 2) the extent of linguistic knowledge needed for reasoning. We acknowledge that, according to current automatic evaluation methods, model performance on ZPR tasks is generally lower than on MRC tasks. To avoid overclaim, in the revised version, we could present the issue of difficulty levels as a more open question for further exploration and discussion.
>
> [1] Translating pro-drop languages with reconstruction models. Wang et al. AAAI 2018.
>
> [2] Coupling context modeling with zero pronoun recovering for document-level natural language generation. Tan et al. EMNLP 2021.
>
> [3] Rethinking Document-level Neural Machine Translation. Sun et al. ACL 2022.
>
> **Q2. What is diagnostic testset? Why not use it to evaluate any models?**
>
> The diagnostic testset is built based on the idea of contrastive testing (whether models can assign higher probability to the candidates with correct discourse marker). As stated in Section 3, we emplyed different ways to create negative sample for MRC, NT and TC tasks. Due to page limitation, we only reported the best-performing models in Table 4 as representatives, facilitating the drawing of some interesting conclusions. We will include all results in the appendix in the revised version for a comprehensive view.
>
> **Q3. About richer linguistic phenomena in literary texts.**
>
> The perspective that literary texts encompass more complex linguistic and cultural knowledge compared to non-literary ones has been substantiated both qualitatively and quantitatively in previous studies [1-2]. This is the primary reason for our decision to construct 9 benchmark tasks focused on the literary domain.
>
> [1] Towards a literary machine translation: The role of referential cohesion. Rob et al. NAACL 2012.
>
> [2] Neural poetry translation. Marjan et al. NAACL 2018.
>
> **Q4. About Novel Translation dataset.**
>
> Yes, tthis dataset comprises 1.4 million sentence pairs, sourced from 5,134 chapters across 152 books.. A significant financial investment was made to annotate this dataset thoroughly. We will make this data available to promote the advancement of research in this field. For details regarding copyright issues, please refer to the response to Reviewer iRfK Q1. For information on our annotation methods, see the response to Reviewer maWf Q3.

---

### Official Review · Reviewer_Q2Uh · 2023-11-01

**Soundness:** 2 fair
**Presentation:** 3 good
**Contribution:** 2 fair
**Rating:** 3
**Confidence:** 4

**Summary:**

The paper presents Disco-Bench---a benchmark of 9 tasks that focus on discourse phenomena. The 9 tasks are constructed with sources from the literature domain and span three groups of tasks: 1) language understanding (Speaker Identification, Zero Pronoun Recovery and Machine Reading comprehension), 2) language translation (Chinese-English Novel Chapter Translation, Classical-Modern Chinese Document-Level translation, Poetry Translation) and 3) language generation (Text expansion, text infilling and text completion). The paper also presents fine-grained diagnostic test sets that evaluate specific phenomena such as repetition, ellipsis, conjunctions ..etc. Additionally, the paper present a large pre-training corpus of long-texts in Chinese and English in the literature domain. Using Disco-Bench, the paper evaluate existing pre-trained models such as BERT, RoBERTa, GPT-3.5 and GPT-4. Results highlight that continued pre-training on in-domain text can boost the accuracy on Disco-bench task and even outperform GPT3.5 as well GPT4

**Strengths:**

1. The paper introduces a benchmark that consists of various tasks that target discourse phenomena.
2. The paper presents results that highlight the limitations of some of the powerful LLMs as well as commonly used smaller pre-trained models.
3. The paper presents a large corpus of Chinese+English text in the literature domain shows that continued pre-training on such corpus can mitigate the limitations of some of the evaluated models.

**Weaknesses:**

1. For pre-trained models evaluation, the benchmark mixes up the Chinese language capabilities of the models under evaluation and the capabilities of such models with discourse phenomena. A model that excels at handling discourse phenomena but is limited with Chinese (e.g., due to lack of Chinese pre-training data) will still perform quite poorly on all tasks of the benchmark.

2. The paper does not provide any qualitative analysis that confirms the conclusions of the comparison results. Also, some of the results in Table 5 need to be explained or at least justified with a reasonable intuition. For example, why does Disco-Bench Pretraining significantly hurst ellipsis handling in the TC task? Also, Disco-Bench pretraining hurts most phenomena in the MRC task. The paper justifies that with "understanding tasks are mostly classification tasks, whose signals may not be sufficient to guide models to learn discourse information" which does not make sense since the continual pre-training phase is independent of the task!

3. Several important details about the benchmark construction and model evaluation are vague:

     3.1. In section 2.1, how was the ground truth constructed for the Speaker Identification task?

     3.2. In section 2.2., how were 45k chapters manually aligned at both the document and sentence-level? What is the definition of a document in the Novel translation task?

     3.3. In Table 4, how were the English-only models (e.g. BERT and RoBERTa) fine-tuned for the tasks in Chinese?

**Questions:**

Please see #3 in Weaknesses above.

---

> ### Author Response · Authors · 2023-11-22
> **Response to Reviewer Q2Uh**
>
> **Q1. The benchmark mixes up the Chinese language capabilities.**
>
> We acknowledge the valid concerns you have raised regarding our dataset design. Our considerations in constructing this dataset are based on the following aspects:
> 1. **Universality of Discourse Knowledge**: Discourse knowledge often shares commonalities across various languages. This implies that merely increasing the data volume for a single language does not necessarily enhance the model's ability to process discourse in that specific language. Instead, the understanding and handling of discourse are largely cross-linguistic. Therefore, our dataset is designed to assess the model's general capability in processing discourse-level information, rather than being limited to a particular language.
> 2. **Importance of Cross-Lingual Capabilities**: In the era of large language models (LLMs), cross-lingual capabilities become increasingly crucial, especially for languages with rich resources like Chinese and English. Our dataset aims to evaluate and foster the development of such abilities, which are essential for building more universally applicable and adaptable LLMs.
> 3. **Diversity of the Dataset**: Both Evaluation Benchmark and Diagnostic Test are designed to encompass both Chinese and English instead of Chinese only.
>
> We believe these considerations are crucial for advancing the field and appreciate the opportunity to clarify our approach.
>
> **Q2. About qualitative analysis.**
>
> Due to page limitation, we did not include a detailed qualitative analysis. However, we recognize the importance of this aspect and will add details in the revised version. For example,
> 1. **Why does Disco-Bench Pretraining significantly hurst ellipsis in TC.** It has been observed that the finetuned model may develop a preference for generating complete sentences with no grammatical omissions, rather than preserving ellipses that require reader inference. This finetuning bias towards producing syntactically complete structures could potentially impact the model's ability to effectively handle ellipsis phenomena.
> 2. **Disco-Bench pretraining hurts Con., Ref., Sub. phenomena in MRC.** In MRC task, such phenomena have a more direct impact on understanding than conjunction, reference, and substitution. For instance, the use of repetition and synonyms can more directly aid the model in capturing the main themes and key information of a text, while ellipsis might play a crucial role in understanding the implied meanings within the text. In the finetuning phase, the primary goal is to optimize the model's performance for a specific task. Consequently, the model may adjust its focus to better align with the demands of the task.
>
> **Q3. About more details.**
>
> 1. **We will add all annotation details in Appendix in the revised version.** Please see Reviewer iRfK Q2  for general annotation process. Specifically, we employed a hybrid method of automatic annotation and manual verification for both the Speaker Identification Task and the Alignment in Novel Translation Task. Specifically, 1) SI, an automated process was used to extract all entities and pronouns from the text based on the context. These extracted elements were then presented to human annotators who selected the correct speaker for each dialogue segment. 2) NT, an automatic alignment tool was initially used to roughly align sentences. This preliminary alignment was subsequently refined through manual corrections.
> 2. **How were the English-only models fine-tuned for the tasks in Chinese?** We have listed all details of pretrained in Appendix A.3. For example, we used BERT-Chinese developped by Hugginface team [1] and chinese-roberta-wwm-ext by Cui et al. [2].
>
> [1] https://huggingface.co/bert-base-chinese.
>
> [2] https://huggingface.co/hfl/chinese-roberta-wwm-ext/tree/main.

---

### Official Review · Reviewer_iRfK · 2023-11-03

**Soundness:** 3 good
**Presentation:** 2 fair
**Contribution:** 3 good
**Rating:** 6
**Confidence:** 2

**Summary:**

The paper presents a benchmark dataset that focuses on
learning/testing discourse phenomena. The dataset includes 9 different
tasks covering "language understanding", translation and text
generation tasks. As well as the main dataset (for training and
development), a small hand-crafted diagnostic test set and a large
unlabeled dataset for language model pretraining were proposed.
Except one, the monolingual datasets are in Chinese
(simplified/classical), translation data is between Chinese variants
The paper also presents results for each task using multiple models.
and English.

**Strengths:**

As noted by the authors, most benchmark datasets focus on
single-sentences (or pairs). The study addresses a relatively less
covered area of benchmark datasets. Paying attention to the
theoretical background (at least during the construction of the
diagnostic test set), and providing examples relevant to coherence and
cohesion is also a strength of the paper.

**Weaknesses:**

The main weakness of the study is unclear description of the data at
many places.

- The paper does not discuss copyright and ethical issues for any of
  the datasets. The source descriptions are also rather vague (e.g.,
  "we crawl 45,134 chapters in 152 books from web fiction websites"
  needs more explicit statements of these websites - perhaps in
  appendix/supplementary material).
- Although some quality assurance (IAA/checks) are reported (Table 2),
  there are unclarities: Were the SI/ZPR/MRC data fully doubly
  annotated? Who were the annotators? Were there any annotation
  guidelines, well-defined procedures?
- Related to above, the description of the "diagnostic test data" is
  also rather terse, and insufficient.

**Questions:**

- It would be nice to specify if there were any issues of text size
  with smaller models trained. Since the texts are long, some models
  (e.g., BERT) may have to truncate the input, or use some other
  mechanism to process the complete data.

Typo/language issues:
- abstract: "We totally evaluate" -> "We evaluate"
- Fig. 1 caption: "propertie" -> "property"
- Table 1: better use full names of the tasks on column 1. There seems
  to be enough space.
- page 3 paragraph on SI: "all speakers are entities, speakers in our
  dataset can also be phrases" "all speakers are names, speakers in
  our dataset can also be phrases describing entities" ?
- page 3 paragraph on ZPR: ", while ZPR considers ..." -> ", ZPR
  considers ..."
- page 4 paragraph on NT: LDC is not a corpus, an organization hosting
  many corpora. If you want to compare your corpus to other parallel
  corpora, you probably need to compare with many available in OPUS.
  Only the subtitle section of the corpus collection does not support your claim.
- For corpora and tools referred to everywhere, prefer citing the
  papers describing them, rather than providing (non-permanent) URL
  references.
- page 6, first paragraph of section 3: "may be not sufficient" ->
  "may not be sufficient"
- Table 3: "Incorrect: She think the Qingshuang is funny." has a
  likely typo in the example: "think" should be "thinks". However,
  this may also point to a systematic error in the corpus. If so, it
  needs to be corrected.
- It is a good practice to report hyperparameters used, but it would
  also be informative to include a statement on how they were determined. Any hyperparameter tuning? Following an earlier example?

**Details Of Ethics Concerns:**

As noted above, there are no statements regarding the copyright and other potential ethical considerations.

---

> ### Author Response · Authors · 2023-11-22
> **Response to Reviewer iRfK**
>
> **Q1. About copyright and ethical issues for any of the datasets.**
>
> Our datasets consist of three types of sources: novel/book, dianji/poetry, composition. We will add the details in appendix:
>
> 1. **novel/book**: we are the copyright owners of these web fictions and books. We make this data available to promote the advancement of research in this field.
> 2. **dianji/poetry**: dianji refers to the Chinese texts which originated before the imperial unification by 221 BC [1] and poetry mainly includes Shi, Ci, Qu, Fu created by 1300 AD [2]. Thus, these are public domain works which are beyond the copyright protection period.
> 3. **composition**: compositions are derived from the reading comprehension section of the Chinese National College Entrance Examination. According to local copyright law, such examination content belongs to "documents of an administrative nature," and therefore is not subject to copyright protection.
>
> [1] https://en.wikipedia.org/wiki/Chinese_classics
>
> [2] https://en.wikipedia.org/wiki/Classical_Chinese_poetry
>
> **Q2. About annotation details of the SI/ZPR/MRC/Diagnostic Test.**
>
> The common anntation process for all datasets is:
>
> A one-week trial annotation phase was conducted for bidding on an enterprise-level annotation platform. The authors answered questions posted by the annotators of these companies and updated annotation guidelines accordingly. The Q&A history is recorded and updated in the formal annotation phase. After evaluating the trial annotations based on accuracy and consistency, we selected the best candidate company with their professional annotators. We ensured that none of the annotators had conflicts of interest, and their annotations were routinely cross-checked for consistency. In terms of compensation, annotators received a fair pay, which aligns closely with the mean hourly wages in U.S..
>
> Some details such as annotator reuqirement varied from nature of different dataset:
> 1. **SI/MRC**: we engaged native Chinese speakers who possess at least a Bachelor's degree. They should have a strong ability of the language  proficiency, contextual understanding and analytical skills.
> 2. **ZPR/diagnostic test**: We employ experts in both source and target languages with Ph.D. degrees in translation and linguistics. These background is essential for accurately labelling nuanced linguistic elements.
> Besides, an exhaustive IRB review was undertaken and finalized before the project's onset. We will add above information and detailed background of each annotator in appendix.
>
> **Q3. About typo/reference issues.**
>
> Accordingly, all typos and missing references will be corrected/citated in our revised version.

---

### Author Response · Authors · 2023-11-22
**General Response to All Reviewers**

We thank all the reviewers for expressing interests and giving insightful comments which will serve to improve the paper considerably. We are encouraged by the positive scores awarded consistently by Reviewers iRfK and maWf. Our gratitude also extends to Reviewer Q2Uh  for their detailed feedback, to which we have responded meticulously.

---

### Meta-Review · Area_Chair_yUpa · 2023-12-13

**Metareview:**

The paper presents a new document-level (cohesion, coherence,...) benchmark, disco-bench, consisting of 9 document-level testbeds within the literature domain.  While the benchmark is valuable, the experimental approaches and attached claims unfortunately leave the door open to serious concerns from the reviewers.  The rebuttal contains mainly promises and weak new motivation rather than concrete new information.

**Justification For Why Not Higher Score:**

While the benchmark is valuable, the experimental approaches and attached claims unfortunately leave the door open to serious concerns from the reviewers.  The rebuttal contains mainly promises and weak new motivation rather than concrete new information.

**Justification For Why Not Lower Score:**

n/a

---

### Decision · Program_Chairs · 2024-01-16

Reject